# Genomic Epidemiology Reveals the Circulation of the Chikungunya Virus East/Central/South African Lineage in Tocantins State, North Brazil

**DOI:** 10.3390/v14102311

**Published:** 2022-10-21

**Authors:** Ueric José Borges de Souza, Raíssa Nunes dos Santos, Marta Giovanetti, Luiz Carlos Junior Alcantara, Jucimária Dantas Galvão, Franciano Dias Pereira Cardoso, Feliph Cássio Sobrinho Brito, Ana Cláudia Franco, Paulo Michel Roehe, Bergmann Morais Ribeiro, Fernando Rosado Spilki, Fabrício Souza Campos

**Affiliations:** 1Bioinformatics and Biotechnology Laboratory, Campus of Gurupi, Federal University of Tocantins, Gurupi 77410-570, Brazil; 2Central Public Health Laboratory of the State of Tocantins, Palmas 77054-970, Brazil; 3Flavivirus Laboratory, Oswaldo Cruz Institute, FIOCRUZ, Rio de Janeiro 21040-360, Brazil; 4Department of Science and Technology for Humans and the Environment, University of Campus Bio-Medico di Roma, 00128 Rome, Italy; 5Virology Laboratory, Department of Microbiology, Immunology, and Parasitology, Institute of Basic Health Sciences, Federal University of Rio Grande do Sul, Porto Alegre 90050-170, Brazil; 6Baculovirus Laboratory, Department of Cell Biology, Institute of Biological Sciences, University of Brasilia, Brasília 70910-900, Brazil; 7Molecular Microbiology Laboratory, Feevale University, Novo Hamburgo 93525-075, Brazil

**Keywords:** CHIKV, ECSA, Brazil, Arbovirus, *Aedes* spp., phylogenetic analysis

## Abstract

The chikungunya virus (CHIKV) is a mosquito-borne virus of the family Togaviridae transmitted to humans by *Aedes* spp. mosquitoes. In Brazil, imported cases have been reported since June 2014 through two independent introductions, one caused by Asian Lineage in Oiapoque, Amapá state, North Region, and another caused by East/Central/South African (ECSA) in Feira de Santana, Bahia state, Northeast Region. Moreover, there is still limited information about the genomic epidemiology of the CHIKV from surveillance studies. The Tocantins state, located in Northern Brazil, reported an increase in the number of CHIKV cases at the end of 2021 and the beginning of 2022. Thus, to better understand the dispersion dynamics of this viral pathogen in the state, we generated 27 near-complete CHIKV genome sequences from four cities, obtained from clinical samples. Our results showed that the newly CHIKV genomes from Tocantins belonged to the ECSA lineage. Phylogenetic reconstruction revealed that Tocantins’ strains formed a single well-supported clade, which appear to be closely related to isolates from the Rio Grande do Norte state (Northeast Brazil) and the Rio de Janeiro state (Southeast Brazil), that experienced an explosive ECSA epidemic between 2016–2019. Mutation analyses showed eleven frequent non-synonymous mutations in the structural and non-structural proteins, indicating the autochthonous transmission of the CHIKV in the state. None of the genomes recovered within the Tocantins samples carry the A226V mutation in the E1 protein associated with increased transmission in *A. albopictus*. The study presented here highlights the importance of continued genomic surveillance to provide information not only on recording mutations along the viral genome but as a molecular surveillance tool to trace virus spread within the country, to predict events of likely occurrence of new infections, and, as such, contribute to an improved public health service.

## 1. Introduction

The chikungunya virus (CHIKV) is a single-strand RNA virus belonging to the Togaviridae family, of the Alphavirus genus, transmitted by *Aedes aegypti* and *A. albopictus* mosquitoes and often causes a polyarthritis polyarthralgia accompanied by fever, muscle pain, rash, and severe joint pain, which may last for months to years [1,2,3,4].

The first identified outbreak of the CHIKV was reported in 1952/53 in the Tanganyika Province, actually Tanzania, East Africa, and since then it has been responsible for important emerging and re-emerging epidemics in several tropical and temperate regions [5]. Phylogenetic analyses reveal that the CHIKV originated in Africa (without a precise localization) and then spread to Asia [6]. Based on genomic diversity, the CHIKV is classified into four distinct genotypes (or lineages): (i) the West African; (ii) the East/Central/South African (ECSA); (iii) the Asian, and (iv) the Indian Ocean Lineage (IOL), which emerged from the ECSA lineage between 2005 and 2006 [7,8].

In the Americas, the first autochthonous CHIKV transmission was reported in December 2013, when the French National Reference Center for Arboviruses confirmed the first local Chikungunya cases on the island of Saint Martin in the Caribbean [9,10]. Sequence analysis confirmed that CHIKV sequences isolated from Saint Martin at the beginning of the outbreak in the Americas belonged to the Asian lineage [9]. By the end of December 2015, approximately one million cases were reported in the Americas, resulting in 71 deaths, and an autochthonous transmission was reported in over 50 countries [11,12].

In Brazil, the local transmission of the Asian lineage was detected for the first time in September 2014 in the Oiapoque city, Amapá state, North Region. Seven days later, a new cluster of CHIKV infections was notified in the city of Feira de Santana, Bahia state, Northeast Region [12,13,14,15]. Since then, the disease has spread across the country, posing a serious threat to public health. Until the epidemiological week 7, Brazil reported 6002 suspected cases (with a rate of incidence of 2.8 cases/100,000). The North Region presents the highest incidence with 7.1 cases/100,000 inhabitants, followed by Central-West with 3.3 cases/100 mi inhabitants [16]. These numbers are smaller than those registered last year but keep the authorities in alarm.

The Tocantins is a Brazilian state located in the North Region, bordering six states, being a corridor for the transmission of pathogens between the North and the Northeast and Central-West Regions [17]. According to the Brazilian Institute of Geography and Statistics (IBGE; https://cidades.ibge.gov.br/brasil/to/panorama, accessed on 1 September 2022), in 2021, the state had an estimated population of 1,607,363 inhabitants, and its human development index was 0.699. The first cases of chikungunya fever in the Tocantins were reported in 2015 and 2017 when the state registered the highest number of notified cases (6669) with an incidence of 200.44 cases per 100,000 inhabitants. The Secretary of Health of Tocantins (SHT) has reported an increase in the number of cases of CHIKV infection at the end of 2021 and the beginning of 2022. Until July 2022, 3794 cases have been notified in the state [18].

Despite several suspected cases reported in the state during the last years, to date, there was still a paucity of complete genomes recovered from cases of chikungunya fever occurring within the state, thus hampering our ability to understand its transmission dynamics. In this study, 27 newly complete CHIKV genome sequences were recovered from autochthonous cases of infection reported in the Tocantins and used to provide a preliminary overview of the likely ways of introduction and circulation of the CHIKV within the state.

## 2. Materials and Methods

### 2.1. Sample Collection and Sequencing

Serum samples from individuals who presented with febrile disease associated with joint pain were collected for molecular diagnostics by the Central Public Health Laboratory of the Tocantins state, North Brazil. Based on resources and time availability, we selected 27 acute-phase serum samples that were RT-qPCR positive for CHIKV with cycle thresholds (Ct) values < 26.9 at the time of diagnosis.

### 2.2. cDNA Synthesis and Whole-Genome Sequencing

Extracted RNA was converted to cDNA by using a Luna Script RT SuperMix (5×) (New England Biolabs, Ipswich, MA, USA). The synthesized cDNAs were used as templates for whole-genome amplification by multiplex PCR using the CHIKV sequencing primers’ scheme (divided into two, separated pools), designed as previously described in [19]. Amplicons from the two primer pools were combined and purified with one volume of AMPure XP beads (Beckman Coulter, Brea, CA, USA), and cleaned-up PCR products’ concentrations were measured using a Qubit dsDNA HS Assay Kit on a Qubit 3.0 fluorometer (ThermoFisher Scientific Corporation, Waltham, MA, USA).

The MinION library preparation was performed using a Ligation Sequencing kit SQK-LSK-109 and Natives Barcoding kits EXP-NBD104 and EXP-NBD114 (Oxford Nanopore, Oxford, UK). The resulting library was loaded on R9.4 Oxford MinION flowcells (FLO-MIN106) and sequenced using a MinION Mk1B device. ONT MinKNOW software was used to collect raw data. High-accuracy base-calling of raw FAST5 files and barcode demultiplexing were performed using Guppy (v6.0.1). Consensus sequences were generated by de novo assembling using Genome Detective [20].

### 2.3. Phylogenetic Analyses

The 27 new CHIKV genome sequences reported in this study were initially assigned to genotypes using the Chikungunya Virus Typing Tool [20].

To compare the phylogenetic relationships of CHIKV genomes, the newly generated sequences were analyzed together with 1156 reference sequences from all genotypes (West Africa, ECSA, and Asian and Caribbean) obtained from the National Center for Biotechnology Information (http://www.ncbi.nlm.nih.gov/ accessed on 5 September 2022) and collected up to July 2022 (Appendix A). Only genome sequences with more than 9000 bases and with recorded dates and sampling locations were selected. Sequences were aligned using MAFFT v7.490 [21] with default settings and manually inspected using AliView v1.28 [22].

The maximum likelihood (ML) phylogeny was reconstructed using IQ-TREE v2.2.0 [23]. The ML analyses were performed under the generalized time-reversible (GTR) model of nucleotide substitution with the empirical base frequencies (+F) plus FreeRate model (+R3), as selected by the ModelFinder software [24] and 1000 replicates of ultrafast bootstrapping (−B 1000) and an SH-aLRT branch test (−alrt 1000). Tree visualization was performed using FigTree v1.4.4 [25].

To infer a time-scaled phylogeny, we extracted sequences of the Brazilian ECSA clade from the phylogenetic tree using a caper R package [26]. A maximum likelihood (ML) phylogeny was reconstructed from the dataset using IQ-TREE2 software under the GTR+G+I nucleotide substitution model with four gamma categories, which was inferred in jModelTest2 v2.1.10 [27] as the best-fitting model. The ML tree was inspected in TempEst v1.5.3 [28] to investigate the temporal signal through a regression analysis of root-to-tip genetic distance against sampling dates. Outlier sequences were removed from the analysis, leaving in the final dataset 186 sequences (Appendix A).

The phylogeny was reconstructed for the Brazilian ECSA clade using Bayesian inference with Markov Chain Monte Carlo (MCMC) sampling, as implemented in the BEAST v1.10.4 package [29]. A stringent model selection analysis using both path-sampling (PS) and stepping stone (SS) procedures was employed to estimate the most appropriate molecular clock model for the Bayesian phylogenetic analysis [30]. For that, we used the strict molecular clock model and the more flexible uncorrelated relaxed molecular clock model with a Bayesian skygrid coalescent model of population size and growth [31,32]. Both SS and PS estimators indicated the uncorrelated relaxed molecular clock was the best-fitted model for the dataset under analysis. Triplicate MCMC runs of 50 million states each were computed, with sampling every 5000 steps. The three independent runs were merged with Log combiner v1.10.4 [29], and the convergence of the MCMC chain was assessed using Tracer v1.7.2 [33]. Maximum clade trees were summarized from the MCMC samples using TreeAnnotator V1.10.4, with 10% removed as burn-in [29]. The MCMC phylogenetic trees were visualized using a ggtree R package [34].

A discrete phylogeographical model [35] was used to reconstruct the virus’ spatial diffusion across the compiled dataset’s sampling locations. Phylogeographic analyses were then performed by applying an asymmetric model of location-transitioning, and location diffusion rates were estimated using the Bayesian stochastic search variable selection (BSSVS) model with a discretization scheme defined as Brazilian states. MCMC was run sufficiently long enough to ensure stationarity and an adequate, effective sample size (ESS) of >200.

## 3. Results

We used, here, a portable MinION sequencer and an amplicon approach to generate 27 partial and near-complete CHIKV genomes (with a coverage range of 74.3–98.4%, and a mean = 87.1%) from the serum samples provided by the Central Public Health Laboratory and collected during 2021 and in January and July of 2022 in the Tocantins state (Table 1). The mean Ct value for the RT-qPCR was 23.0 (range: 16.2 to 26.9). These samples were in the majority of male patients (n = 14; 51.9%), and the mean patient age was 34.2 years (Table 1). Most of the isolates (n = 20; 74.1%) belonged to patients that resided in the municipality of Palmas, the capital of the Tocantins state. The remaining seven samples were from Lavandeira (*n* = 2; 7.4%), and two neighboring municipalities of the capital, Paraíso do Tocantins (*n* = 2; 7.4%) and Porto Nacional (*n* = 3; 11.1%) (Figure 1).

### 3.1. Phylogenetic Analysis

The initial maximum likelihood phylogenetic tree was constructed using the dataset containing 1156 sequences from the four distinct genotypes plus the sequences recovered here. This analysis revealed that all 27 sequences belonged to the ECSA lineage and clustered with other Brazilian strains from previous outbreaks reported in other geographic regions (Appendix A). These findings were also confirmed by the analysis performed with the Chikungunya Virus Typing Tool.

The temporal analysis of the Brazilian ECSA sequences showed a sufficient temporal signal (with R2 = 0.78 and a correlation coefficient of 0.88; Appendix A) between the genetic divergence from root to tip against the sampling dates, supporting the use of temporal calibration directly from the sequences. The evolutionary rate of the ECSA lineage estimated in this study was 2.64 × 10^−3^ substitutions per site, per year (95% highest posterior density interval-HDP: 1.96 × 10^−3^ to 3.28 × 10^−3^ substitutions per site, per year). The time of the most recent common ancestor (TMRCA) of the Tocantins clade was estimated to be around March 2021 (95%: HDP: November 2020 to May 2021).

The Bayesian phylogeny showed that the CHIKV isolates from the Tocantins state collected between 2021 and 2022 and formed a single, well-supported clade (posterior probability = 1.0), which was closely related to the isolates from the 2019 CHIKV epidemic in both Rio de Janeiro and Rio Grande do Norte states (Figure 2). The BSSVS procedure identified well-supported rates of diffusion from Rio de Janeiro to Tocantins (Bayes Factor: 29.9; Posterior Probability: 0.77). This suggests that the outbreak in Tocantins was caused by an introduction event from Rio de Janeiro, which appears to have been mediated also by the Northeast Region, which has played an important role in the introduction and establishment of the ECSA lineage in Brazil.

### 3.2. Single Nucleotide Polymorphisms (SNPs) Analysis

The genomic analysis revealed 110 different mutations in the 27 sequences obtained in this study (Appendix A). Of the total mutations detected, 76 (69.1%) were synonymous and 34 were missense (30.9%) (Table 2). Highly frequent mutations (found in at least twelve genomes or ≥44.4% of the genomes) were found in 42 genome positions. Of these, 73.8% (31) were synonymous and 26.2% (11) were missense. Among the highly frequent missense mutations, six were located in the nonstructural polyprotein sequence (NSP2: P352M, M466L, and A545S; NSP3: A388T and L434P; NSP4: A481D) and five in the structural protein (E1: A98T, K211T, V269M, and A305T; E2: L248F) (Table 2).

## 4. Discussion

Previous phylogenetic data are not available to show the circulation of autochthonous CHIKV in the Tocantins. Our phylogenetic analysis placed the genomes to the ECSA genotype. This lineage has been reported as predominant in the Southeast and Northeast Region of Brazil, respectively [12,36,37,38,39]. Naveca et al. [40] reported the introduction of the ECSA lineage in Roraima in the North Region of Brazil, and de Castro et al. reported the ESCA circulation in the Paraná state in the South Region of Brazil. Taken together, this makes it clear that the ECSA lineage is prevalent and spread across the country [41].

The analysis of the full or nearly full genomes maximizes the usefulness of molecular clock models to infer time-scaled phylogenetic trees [28]. The Bayesian phylogeny demonstrates that all TO-CHIKV genomes reported here belong to a distinct clade from those previously identified in Brazil and were phylogenetically close to the 2019-Rio Grande do Norte [42] and 2019-Rio de Janeiro cluster [36]. We also found evidence of dispersal from Rio de Janeiro to the Tocantins, mediated from the Northeast Region. The strategic location of the Tocantins state was previously demonstrated as a transmission virus corridor between the North and South Regions of Brazil [17]. We agreed that the dispersion of the CHIKV is influenced by multiple factors, including the geographic parameters and traffic metrics that contribute to the vector transport. In addition, population density improves the chances of encountering a susceptible individual.

Genomic analysis revealed eleven highly frequent missense mutations in the twenty-seven TO-CHIKV genomes: six of those were placed in the nonstructural polyprotein and five in the structural polyprotein. This set of mutations characterizes the profile of sequences evidencing the autochthonous transmission of the CHIKV in Tocantins. Mutations in the genome could impact the viral fitness of these lineages in different vectors [3]. The presence of the T98A variant on the envelope gene (E1) of the CHIKV was associated with the enhancement of the vector-adaptability of *A. albopictus*. The substitution of E1-A226V was reported to suffer and exert strong epistatic effects, blocking the ability of most of the ECSA and endemic Asian strains to *A. albopictus* adaptability [6,43]. However, strains whose epidemiologic success contained E-211T and E1-98A residues were not affected by these interactions [6].

Studies showing genomic differences among CHIKV circulation are important to determine the changes or neutral occurrences in the viral genome. The evolution of the CHIKV with novel mutations has proven to be the epidemic potential for viral re-emergence due to altered vector specificity and infectivity. Additionally, the K211E variant in the E1 gene and V264A in the E2 gene leads to an increase in viral dissemination and transmission for *A. aegypti* but not for *A. albopictus* [44]. The E1 adaptative mutation, E1-A226V, enhances CHIKV replication in *A. albobictus*. Furthermore, other variants in genes E1 and E2 increase, enhance, or promote an epistatic effect in vector dissemination. Our study detected sequences without substitutions previously described, allowing virus transmission by different species of mosquitoes [44,45].

Moreover, other mutations already found among Brazilian samples in the envelope protein (E1: A98T, K211T, V269M, and A305T) were detected in this study. The impact of these mutations on CHIKV fitness stays under investigation [39,46]. The K211T amino acid substitution in the E1 gene was described for the first time in Rio de Janeiro and was found in all of the samples analyzed without the A226V mutation [39]. This reinforces the need for further studies to better understand these mutations and their consequences on vector fitness and the human immune system.

Several adaptive mutations enhance CHIKV transmissibility by different mosquito species. The appearance of those adaptive mutations has led to large-scale CHIKV outbreaks in Asia, Africa, Europe, and Brazil [47]. The only method to control the CHIKV is to decrease the exposure between people and mosquitoes. In remote and extensive nature or forest habitats, vector control is a challenge. However, the reduction of endemic/epidemic transmissions is feasible in regions where resources are designated to minimize the proliferation of the vectors. The largely sympatric distribution of the vectors, as well as the increasing cocirculation of different lineages, were important factors to consider in tracking/preventing the dissemination of mosquitoes and genomic surveillance [48].

In this study, we have provided the CHIKV genomic epidemiology showing the dynamic spread of the CHIKV ECSA lineage from symptomatic cases reported in the Tocantins state. Our research enhances the number of CHIKV genome sequences available, allowing us to learn more about the evolution and genomic diversity of the ECSA lineage. In addition, this emphasizes genomic surveillance strategies in order to track viral adaptations. Several conditions, such as the high density of vector mosquitoes, high population density, poor income, and insecure sanitary conditions, as well as significant air connectivity and land transport between different Brazilian regions with a constant movement of people, may contribute to the modeling of this complex viral transmission scenario.

Some limitations of this study involve the small number of samples analyzed in relation to the total number of cases. In addition, there is no frequent monitoring in all of the Brazilian states, which makes it difficult to establish the real dispersion route of the CHIKV in Brazil. Moreover, it is important to specify that we do not designate if this is a new clade in the state or even if this clade already exists in other Brazilian regions. The principal limitation is that there were no sequences from the state prior to this study. However, the role of the most populous regions, such as the state of Rio de Janeiro, as an amplifier of the virus and disseminator to other Brazilian regions appears to be clear. This makes monitoring through genomic surveillance more important and should be something to be implemented and used in all of the Brazilian states to combat arboviruses, especially the CHIKV, the target of the present study.

## Figures and Tables

**Figure 1 viruses-14-02311-f001:**
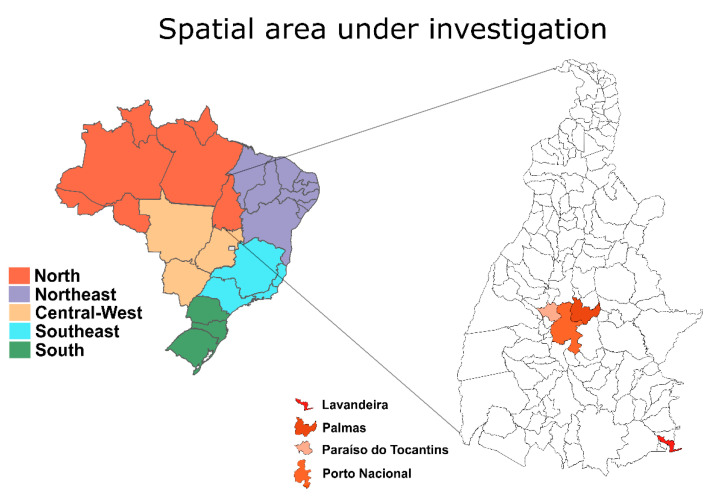
Political map (MOU1) of Brazilian regions with highlights on the Tocantins state, showing locations of the origin of the patients in which the CHIKV genomes reported in this study were collected. The figure was generated using R v4.1.2 with the following packages: ggplot2, geojsonio, sf, broom, and dplyr, and manually edited using Inkscape (GNU GPL v1.2; See http://www.inkscape.org, accessed on 12 September 2022).

**Figure 2 viruses-14-02311-f002:**
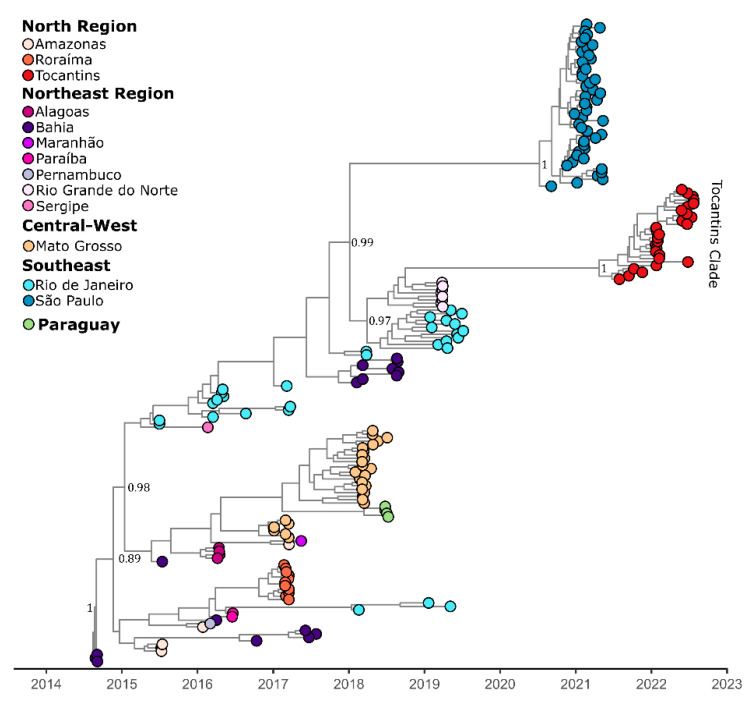
Time-scaled phylogenetic tree of 186 complete and near-complete CHIKV genome sequences from the ECSA genotype sampled in Brazil and Paraguay. Colors represent different sampling locations according to the legend on the left of the tree. Genomes reported in the presented study are colored in red.

**Table 1 viruses-14-02311-t001:** Epidemiological data associated with sequenced samples were analyzed in this study.

Sample ID	Accession ID	Cycle Threshold	Coverage	Depth of Coverage	Host	State	Municipality	Collection Date	Sex	Age
TO-UFT-245	ON586954	25.2	98.2	3588.8	Human	Tocantins	Lavandeira	2021-09-14	Male	17
TO-UFT-7124	ON586955	24.3	98.4	3448.3	Human	Tocantins	Paraíso do Tocantins	2021-07-30	Male	39
TO-UFT-252	ON586956	22.1	98.4	4059.3	Human	Tocantins	Lavandeira	2021-10-08	Female	13
TO-UFT-5070	ON586957	21.3	98.4	4338.4	Human	Tocantins	Palmas	2021-11-18	Male	56
TO-UFT-22529	ON586958	19.7	74.3	1035.7	Human	Tocantins	Palmas	2022-01-21	Female	53
TO-UFT-33522	ON586959	21.6	91.5	1499.3	Human	Tocantins	Palmas	2022-01-22	Male	51
TO-UFT-67522	ON586960	18.9	77.0	1088.9	Human	Tocantins	Palmas	2022-01-24	Female	27
TO-UFT-5531	ON586961	16.2	94.4	3437.5	Human	Tocantins	Palmas	2022-01-25	Female	54
TO-UFT-18531	ON586962	20.7	79.1	910.8	Human	Tocantins	Palmas	2022-01-26	Female	75
TO-UFT-32531	ON586963	21.2	87.8	1633.8	Human	Tocantins	Palmas	2022-01-26	Male	4
TO-UFT-50531	ON586964	22.4	83.5	959.7	Human	Tocantins	Palmas	2022-01-31	Female	48
TO-UFT-52531	ON586965	20.4	90.9	1686.3	Human	Tocantins	Palmas	2022-01-31	Male	18
TO-UFT-72569	ON586966	23.6	84.0	1151.4	Human	Tocantins	Palmas	2022-02-07	Male	36
TO-UFT-86569	ON586967	25.2	94.0	3830.9	Human	Tocantins	Palmas	2022-02-07	Female	53
TO-UFT-64569	ON586968	26.9	94.2	1795.7	Human	Tocantins	Palmas	2022-02-09	Female	38
TO-UFT-9217	OP485445	26.7	84.9	2884.7	Human	Tocantins	Palmas	2022-05-27	Female	26
TO-UFT-9317	OP485446	24.0	81.5	2897.4	Human	Tocantins	Palmas	2022-05-26	Female	66
TO-UFT-2017	OP485447	25.9	83.7	3385.3	Human	Tocantins	Porto Nacional	2022-05-29	Female	9
TO-UFT-4345	OP485448	24.7	85.0	2513.0	Human	Tocantins	Palmas	2022-06-21	Male	27
TO-UFT-8545	OP485449	23.9	86.3	2618.0	Human	Tocantins	Palmas	2022-06-22	Female	8
TO-UFT-3045	OP485450	24.1	84.2	3209.1	Human	Tocantins	Palmas	2022-06-24	Male	21
TO-UFT-6145	OP485451	24.3	86.1	3840.0	Human	Tocantins	Palmas	2022-06-24	Female	9
TO-UFT-4945	OP485452	23.1	85.0	3269.7	Human	Tocantins	Palmas	2022-06-26	Male	37
TO-UFT-2447	OP485453	23.8	81.7	2791.1	Human	Tocantins	Palmas	2022-07-14	Male	25
TO-UFT-2645	OP485454	23.5	86.6	2897.2	Human	Tocantins	Porto Nacional	2022-07-20	Male	26
TO-UFT-6747	OP485455	25.8	84.1	2603.7	Human	Tocantins	Paraíso do Tocantins	2022-07-23	Male	41
TO-UFT-447	OP485444	22.5	79.6	3968.6	Human	Tocantins	Porto Nacional	2022-07-24	Male	46

**Table 2 viruses-14-02311-t002:** Missense mutations found in the newly sequenced genomes of CHIKV from Tocantins, between 2021 and 2022, reported in this study. Showing only mutations observed in at least two sequenced genomes.

	Nonstructural Protein (NSP)															Structural Protein (SP)							
Protein	NSP1		NSP2							NSP3				NSP4		E2		E1					
Sites (Protein)	149	256	352	466	470	540	545	565	645	284	388	434	464	39	481	248	391	35	98	211	269	305	366
KP164568 aa *	Ile	Lys	Pro	Met	Thr	Val	Ala	Ala	Val	Ile	Ala	Leu	Ala	Lys	Ala	Leu	Met	Ser	Ala	Lys	Val	Ala	Arg
TO-UFT-7124	Val		Ala	Leu			Ser					Pro			Asp					Thr	Met	Thr	
TO-UFT-245	Val		Ala	Leu			Ser					Pro			Asp	Phe				Thr	Met	Thr	
TO-UFT-252	Val		Ala	Leu			Ser					Pro			Asp					Thr	Met	Thr	
TO-UFT-5070	Val		Ala	Leu			Ser					Pro			Asp					Thr	Met	Thr	
TO-UFT-22529			Ala				Ser	Val	Ala		Thr	Pro			Asp					Thr	Met		
TO-UFT-33522	Val		Ala	Leu			Ser				Thr	Pro			Asp				Thr	Thr	Met	Thr	
TO-UFT-67522			Ala	Leu			Ser								Asp				Thr	Thr	Met	Thr	
TO-UFT-5531	Val		Ala	Leu	Ile		Ser				Thr	Pro			Asp	Phe				Thr	Met	Thr	
TO-UFT-18531			Ala	Leu			Ser		Ala						Asp	Phe			Thr	Thr	Met	Thr	
TO-UFT-32531			Ala	Leu			Ser				Thr	Pro			Asp				Thr	Thr	Met	Thr	
TO-UFT-50531			Ala	Leu		Met	Ser		Ala		Thr	Pro			Asp			Leu	Thr	Thr	Met	Thr	
TO-UFT-52531			Ala	Leu		Met	Ser		Ala		Thr	Pro			Asp	Phe		Leu	Thr	Thr	Met	Thr	
TO-UFT-72569			Ala	Leu			Ser		Ala		Thr	Pro		Arg	Asp				Thr	Thr	Met	Thr	
TO-UFT-86569	Val		Ala	Leu			Ser				Thr	Pro			Asp	Phe				Thr	Met	Thr	
TO-UFT-64569	Val		Ala	Leu			Ser				Thr	Pro			Asp	Phe				Thr	Met	Thr	
TO-UFT-9217			Ala	Leu			Ser				Thr	Pro			Asp	Phe			Thr	Thr	Met	Thr	
TO-UFT-9317			Ala	Leu			Ser	Val	Ala		Thr	Pro	Val		Asp	Phe			Thr	Thr	Met	Thr	Leu
TO-UFT-2017			Ala	Leu			Ser				Thr	Pro		Arg	Asp	Phe			Thr	Thr	Met	Thr	
TO-UFT-4345		Arg	Ala	Leu			Ser				Thr	Pro			Asp	Phe	Ile		Thr	Thr	Met	Thr	
TO-UFT-8545			Ala	Leu			Ser				Thr	Pro			Asp	Phe	Ile		Thr	Thr	Met	Thr	
TO-UFT-3045			Ala	Leu			Ser				Thr	Pro			Asp	Phe			Thr	Thr	Met	Thr	
TO-UFT-6145			Ala	Leu			Ser				Thr	Pro			Asp	Phe			Thr	Thr	Met	Thr	
TO-UFT-4945			Ala	Leu	Ile		Ser		Ala		Thr	Pro			Asp	Phe				Thr	Met	Thr	
TO-UFT-2447		Arg	Ala	Leu			Ser				Thr	Pro			Asp	Phe	Ile		Thr	Thr	Met	Thr	
TO-UFT-2645			Ala	Leu			Ser			Thr	Thr	Pro			Asp				Thr	Thr	Met	Thr	
TO-UFT-6747			Ala	Leu			Ser	Val			Thr	Pro	Val		Asp	Phe			Thr	Thr	Met	Thr	Leu
TO-UFT-447			Ala	Leu			Ser		Ala	Thr	Thr	Pro			Asp	Phe			Thr	Thr	Met	Thr	

* Amino acids (aa) abbreviations: Ala: Alanine; Arg: Arginine; Asn: Asparagine; Asp: Aspartic acid; Glu: Glutamic acid; Ile: Isoleucine; Leu: Leucine; Lys: Lysine; Met: Methionine; Phe: Phenylalanine; Pro: Proline; Ser: Serine; Thr: Threonine; Val: Valine. Protein (amino acid numbers): Nonstructural protein (2474), NSP2 (798), NSP3 (524), NSP4 (611), Structural protein (1248), E2 (423), 6k (61), and E1 (439).

## Data Availability

The authors declare that all data supporting the findings of this study are available within the paper. Analysis scripts and the dataset used are available at: https://github.com/Ueric/Chikungunya-ECSA-lineage-Tocantins-state.git (accessed on 1 September 2022).

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
