# Peer review of "Genomic Epidemiology Reveals the Circulation of the Chikungunya Virus East/Central/South African Lineage in Tocantins State, North Brazil"

_viruses, 2022, doi:10.3390/v14102311_

Round 1
Reviewer 1 Report (Previous Reviewer 3)
none
Reviewer 2 Report (Previous Reviewer 1)
I am happy with the changes in the manuscript as well as with the addressing of the questions proposed by reviewers.
This manuscript is a resubmission of an earlier submission. The following is a list of the peer review reports and author responses from that submission.
Round 1
Reviewer 1 Report
Borges de Souza et al provide genomic epidemiology in a region of Northern Brazil showing the dynamic spread of CHIKV. They analyse 15 CHIKV genomic sequences produced from patient serum samples and looked at their diversity and genetic sequences based in the context of available local CHIKV genomic sequences. They concluded that the ECSA lineage is prevalent and has become endemic across the country. Their research also highlights how well connected populous urban centres act as amplification hubs for the virus and that it spreads to other regions of the country from there.
The manuscript is interesting and through some light the in spread dynamics of a newly introduced arbovirus into an area where it is becoming endemic. The manuscript is well written and I enjoyed reading it. The methods are sound, and the results are well presented. Conclusions align with what’s shown on the results. The discussion was also interesting, context of mutations is well explained.
Limitations, like the small number of samples are addressed, but that is due to the lack of monitoring of febrile illnesses in the local health system. It would have been nice to complement this data with local mosquito isolated CHIKV sequences since they are probably easier to access that vaeremic serum samples. Timing for extracting those serum samples is very limited while the patient has enough CHIKV in their peripheral blood.
I would like the author's to address the following questions:
- How does this correlate to local mosquito extracted CHIKV sequences?
- Are there other local Aedes mosquito species that transmit CHIKV other than albopictus?
- Do the authors think that the virus spreads from urban centres in Brazil (Sao Paulo/Rio de Janeiro) to regional areas through humans traveling or through mosquitoes being accidentally moved?
- Do you see any convergent evolution with some other CHIKV sequences around recently established endemic areas?
Reviewer 2 Report
It is my impression that this paper is not making a big contribution in the CHIKV research area; it is focused on an area of Brazil and perhaps more suitable for a Brazilian journal, like previous paper published on the Brazilian journal of Microbiology (reference 40).
A weakness is the small number of samples analysed (15) out of the total number of cases (2056), therefore less than 1 %. However, critically, understanding CHIKV in Tocantins does not appear to have any broader consequences for the field. Hard to see how it would have public health implications or provide new insights; as might be exemplified by, for instance; https://www.biorxiv.org/content/10.1101/2022.03.16.484685v1.abstract; or International Journal of Infectious Diseases 2021; 113, 65-73.
Furthermore, there is also a similar paper from another Brazilian group (Following in the Footsteps of the Chikungunya Virus in Brazil: The First Autochthonous Cases in Amapá in 2014 and Its Emergence in Rio de Janeiro during 2016 ). They analysed bigger number of samples.
Introduction; the main symptom might best be described as polyarthritis polyarthralgia, which may be accompanied by fever rash etc (Nature Reviews Rheumatology 15, 597–611 (2019)
The discussion does not make a compelling case that the information generated is of value or interest or lead to any new understanding(s), it states many things that could equally go into the introduction. Obtaining sequence data is in as of itself valuble, but surely some story of broader interest would need to emerge before this becomes suitable for publication for Viruses?
Reviewer 3 Report
Manuscript ID: viruses-1829278
Title: Genomic epidemiology reveals the circulation of the chikungunya virus East-Central South African lineage in Tocantins state, North Brazil
Authors: Ueric José Borges de Souza *, Raíssa Nunes dos Santos, Marta Giovanetti, Luiz Carlos Júnior Alcantara, Jucimária Dantas Galvão, Franciano Dias Pereira Cardoso, Feliph Cássio Sobrinho Brito, Ana Cláudia Franco, Paulo Michel Roehe, Bergmann Moraes Ribeiro, Fernando R. Spilki, Fabricio S. Campos
The authors sequences 15 complete chikungunya virus (CHIKV) genomes from theTocantins state, located in North Brazil, to better understand the evolution dynamics of this viral pathogen in this state. Their data showed that the new CHIKV genomes belong to the ECSA lineage. Phylogenetic reconstruction revealed that Tocantins’ strains formed a single clade, which appear to be closely related with isolates from Rio Grande do Norte state (Northeast Brazil) and Rio de Janeiro state (Southeast Brazil), that experienced an explosive ECSA epidemic between 2016–2019. The new viruses had eleven frequent non-synonymous mutations in structural and non-structural proteins, without the A226V mutation in the E1 protein, which is associated with increased transmission in A. albopictus. The study contributes very well to the general analysis of CHIKV evolution.
The study is very clearly written and the data support the conclusion. There is no criticism to the study.
